# Ecological status of a freshwater tectonic lake of the indo-burmese province: Implications for livelihood development

Arup Kumar Hazarika[1☯], Unmilan Kalita[2☯], Rev. George Michael[3‡], Saroj Panthi[4]*, Dulumoni Das[5‡]

1 Department of Zoology, Cotton University, Guwahati, India, 2 Department of Economics, Cotton University, Guwahati, India, 3 Former Professor and Vice-Chancellor, North Eastern Hill University, Meghalaya, India, 4 Ministry of Industry, Tourism, Forest and Environment, Gandaki Province, Pokhara, Nepal, 5 Department of Statistics, Cotton University, Guwahati, India

☯ These authors contributed equally to this work.
‡ RGM and DD also contributed equally to this work.
* mountsaroj@gmail.com

**Data Availability Statement:** Data are available at: https://doi.org/10.6084/m9.figshare.13079675.

**Funding:** The author(s) received no specific funding for this work.

## Abstract

Tectonic lakes are among the most geologically fascinating and environmentally versatile hydrobiological systems found on the earth's surface. We conducted a study on the limnology of Tasek Lake, a tectonic lake located in the Indo-Burma Province of the South Asian region. Physico-chemical parameters of the lake's water along with its plankton were considered for the study. Their relationship was analysed by understanding their seasonal variations and through linear regression models. The water quality index (WQI), plankton diversity indices and canonical correspondence analysis (CCA) were computed. The ichthyofaunal diversity was also studied to get an insight into the lake's fishery potential. A preliminary assessment on the economic feasibility of converting Tasek Lake into a fishery was also completed. Results indicate moderate eutrophication in the lake and the plankton population is observed to be rich and abundant. The WQI value confirms the water to be of "very poor" quality. The CCA was done to analyze the relationships of physico-chemical parameters with months and seasons, and the relation between seasons and plankton assemblages. Results corroborate the results of WQI. Identified fish population suggest ample fishery potential of the lake. The economic assessment reveals that in order to maintain the ecological sustainability of the lake, it should be transformed into a recreational fishery, following a catch-and-release model. The study calls for urgent restoration of the lake so that not only its pristine ecology is survived but also its fishery potential is sustainably harnessed and local livelihood is improved.

## Introduction

Formation of lakes is an evidence of the abundant geological activity that constantly shapes and re-shapes the physical structure of Earth. As a result of vertical or horizontal disturbances deep within the Earth's crust, tectonic lakes are formed [1]. Such lakes are found all over the

**Competing interests:** The authors have declared that no competing interests exist.

world, though not frequently as glacial or oxbow lakes. Tasek Lake, located in the State of Meghalaya, India, is one of the few and most important tectonic lakes situated in the South Asian region. It was created following the massive 1897 Assam earthquake which had a magnitude of 8. $M_w$ [2] and Meghalaya's state capital, Shillong, was the epicentre. This tectonically active region was originally a part of the Indian peninsular plateau which is characterised with rich and extensive biodiversity unique to the Indo-Burma biodiversity hotspot [3].

A preliminary survey through reputed international online databases show that global literature on limnological reconnaissance of tectonic lakes, especially freshwater, is scant. In India, limnological studies have been done majorly on the Wular Lake [4, 5], Chandubi Lake [6] and other small lakes located deep in the Himalayan region [7]. This scantiness is seen more so in case of India's biodiversity rich North-Eastern region, of which Tasek Lake is a critical part. Notable limnological studies on Tasek Lake have been previously conducted [8, 9] and [10]. Nevertheless, growing urbanisation and tourism in the Meghalayan region, it has become imperative to investigate whether the aquatic ecology of Tasek Lake is close to being pristine.

This study aims to perform a limnological assessment of the Tasek Lake in relation with the physico-chemical parameters of its water, plankton abundance and diversity by determining its water quality and diversity indices. To analyze the relationship of physico-chemical parameters with months and seasons, and relationship between seasons and plankton assemblages, Canonical Correspondence Analysis (CCA) was performed. The ichthyofaunal diversity has also been studied to get an insight into the lake's fishery potential. The study is accompanied by a discussion on the economic benefits of pisciculture in the lake.

## Materials and method

### Study area

Tasek Lake is one of the few significant tectonic freshwater lakes found in India. It is located at $90^0 11'E$, $25^0 34'N$ in the State of Meghalaya, India (Fig 1). Three sampling sites were chosen from the lake- S1 (25˚37'46.1"N 90˚06'20.5"E), S2 (25˚37'47.8"N 90˚06'19.6"E) and S3 (25˚ 37'49.2"N 90˚06'22.9"E). Standing at an elevation of approximately 600 metres (1968.5 feet) above sea level, the lake has surface area of 11.66 Ha. It has a mean depth of 2.8 metres and a maximum depth of 6.75 metres. The nearest urban centres are Tura (West Garo Hills district) and William Nagar (East Garo Hills district) which are located at a distance of 131 km and 32 km respectively from the lake. Originally a hillock, the Tasek Lake was formed as a result of the great earthquake of 1897 [11]. Sal (*Shorea robusta*) trees prodding out of the water surface stands as its evidence. The lake is bounded by a thick forest on the northern and western side while a hillock covers the southern bank. A small water body, Chitmarang Lake, adjoins its eastern side, both of which are connected by a narrow stream. The climate of the area is neither too warm in summer ($19^0$–$22^0$ C) nor too cold ($5^0$–$7^0$ C) in winter. The average annual rainfall is about 1,150 cm.

### Collection of samples

The sampling of water was carried out for four seasons throughout the period 2019 (February)– 2020 (January). As per [12], the seasons have been climatologically classified as pre-monsoon (March, April and May), monsoon (June, July, August and September), retreating monsoon (October and November) and winter (December, January and February). Three sampling sites were selected from the lake (Fig 1).

Monthly sampling was undertaken collected during early hours of the day (0500 hrs—0900 hrs) in triplicate in acid-washed, dried, polyethylene bottles. The samples were collected from a depth of 10 to 20 cm. After collection, the collected samples were thoroughly mixed and

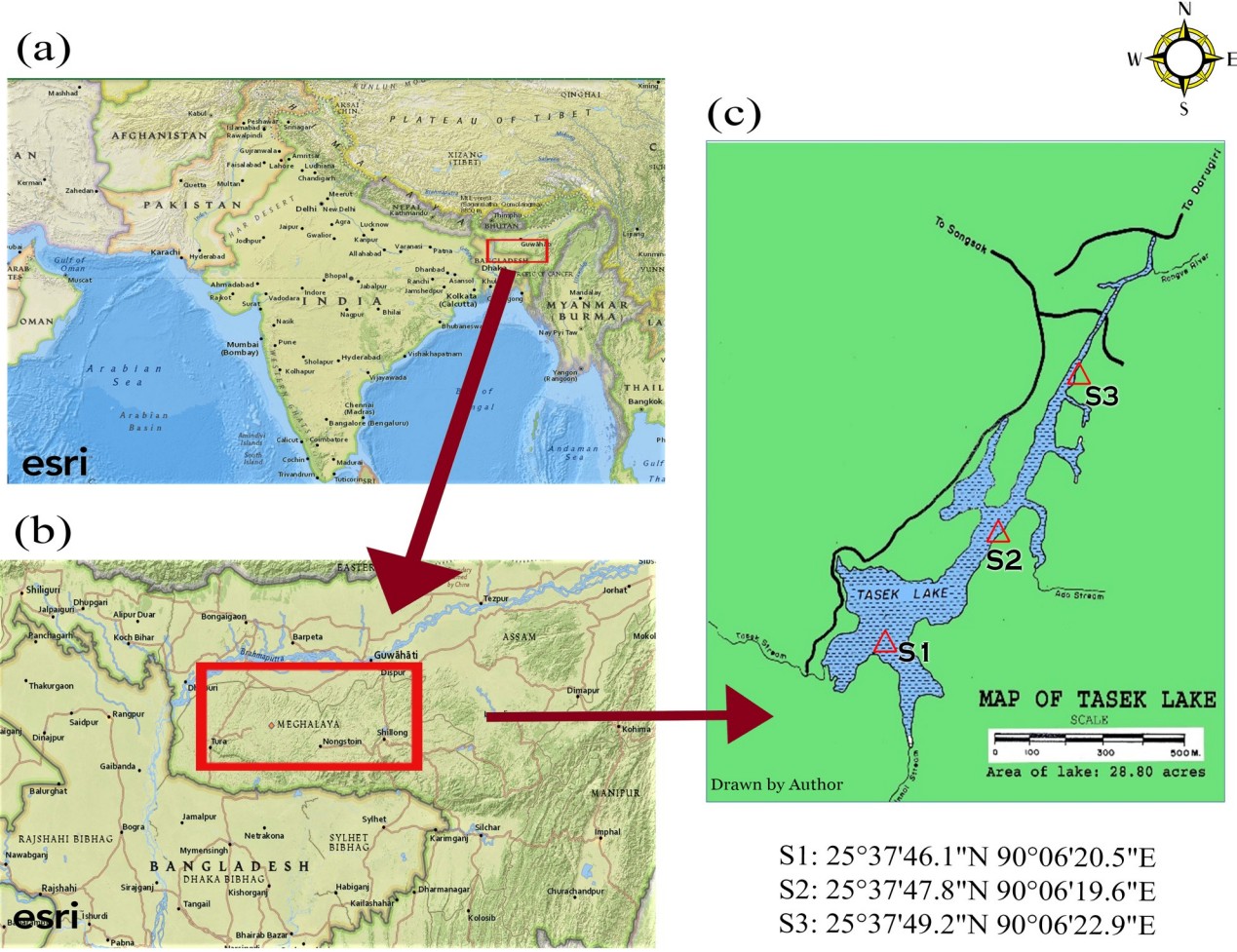

**Fig 1.** Study area (a- Map of India; b: Map of Meghalaya; c: Map of Tasek Lake).

S1: 25°37'46.1"N 90°06'20.5"E
S2: 25°37'47.8"N 90°06'19.6"E
S3: 25°37'49.2"N 90°06'22.9"E

stored in iceboxes for impeding biochemical reactions. Double distilled water was used to create the standard reagents. Overall, the sampling was carried out as per American Public Health Association [13] methods.

Permissions were obtained from Office of the Divisional Forest Officer (DFO), East and North Garo Hills Division, Government of Meghalaya, and the Nokma of Napak A.king Land, Council of Nokmas, Tura, Meghalaya, for carrying out sample collection in the Tasek Lake. All sampling procedures and/or experimental manipulations were reviewed as part of obtaining the field permit. Further, the Committee for Control and Supervision of Experiments on Animals (CCSEA) of the Centre for Environment, Education and Economic Development (CEEED), India, provided us with permission for sampling of fishes and do the required vertebrate work.

## Physico-chemical analysis

Fifteen physico-chemical parameters were considered for water quality analysis of Tasek Lake. They are, namely, pH, water temperature (Temp.), conductivity (C), transparency (T), hardness (H), alkalinity (A), chloride (Cl), magnesium (Mg), phosphate (PO), nitrate (NI), iron (Fe), free carbon dioxide ($CO_2$), dissolved oxygen (DO), biological oxygen demand (BOD)

and chemical oxygen demand (COD). Water temperature was measured immediately after sample collection using a centigrade thermometer. The pH and conductivity were measured by PC Stester 35. A 20 cm Secchi disc was used to measure the transparency. Alkalinity and free $CO_2$ were measured through titration while DO, BOD and COD were gauged using Winkler method. Concentration of the elements was determined through standard APHA [13] methods.

## Statistical analysis

The Water Quality Index (WQI) has been computed in three stages [14]. In the first stage, weights (0–5) were assigned to each of the fifteen parameters according to their relative importance in the overall water quality (drinking standards). Highest weight was assigned to NI while the lowest weight was assigned to Mg given their relative importance in conventional water quality analysis. Following this, the relative weight ($W_i$) was calculated using the Eq (1).

$$W_i = \frac{w_i}{\sum w_i} \tag{1}$$

where, $w_i$ is the weight assigned to each parameter, and i = 1. . .. n; n is the total number of parameters.

The third stage comprises of computing a quality rating scale ($Q_i$) for each parameter as denoted by Eq (2).

$$Q_i = C_i/S_i \times 100 \tag{2}$$

where, $C_i$ is the concentration of a particular parameter in each water sample, $S_i$ is the permissible limit/standard according to an internationally accredited organisation. We have considered the Bureau of Indian Standards (BIS) guidelines for this study. Finally, Eq (3) denotes the computation of WQI. The resultant WQI values are distinguished into five types namely, excellent, good, poor, very poor, and unsuitable for drinking.

$$WQI = \sum (W_i Q_i) \tag{3}$$

The Canonical Correspondence Analysis (CCA) was performed via SPSS (v.23) and PAST v3.2. CCA was used to determine the relationship of physico-chemical parameters with months and seasons along with relationship between months and plankton assemblages distinctly to get a comprehensive idea of the overall ecological status of the tectonic lake.

## Sampling of ichthyofauna

Collection of fishes through timely monitoring and frequent netting during the study period. Since there are submerged trees, stones and large boulders in the lake, assistance of local people was taken in order to collect fishes through indigenous techniques. After collection, 4% formalin was used to preserve the fishes.

## Plankton abundance and diversity indices

Plankton sampling from Tasek Lake was completed for all the four seasons. The procedure involved filtering 20 litres of water through a plankton net (standard grade No. 25 and 60 microns mesh size) made of bolting silk. Sampling was done in early hours of the day. Preservation was done in 4% formaldehyde after which they were kept for 24 hours in a 100 ml borosil graduated centrifuge tube. This allowed for settling of the organic elements. At this stage, the volume was recorded. After macro-plankton were removed, samples were stirred so that the organisms were distributed evenly. For counting, the plankton were transferred into a

Sedgwick Rafter counting cell through a narrow-mouthed pipette. Identification was done as per [15] and [16].

For determining the effect of physico-chemical parameters on plankton abundance, variations of the following linear regression model were used.

$$Y = \beta_0 + \beta_1 X_1 + \beta_2 X_2 + \cdots + \beta_n X_n + \mu \tag{4}$$

where, $\beta$, $\beta 1 \ldots \beta n$ = constants, Y = Plankton abundance (U/l), $X_1$ = pH, $X_2$ = Water temperature (°C), $X_3$ = Conductivity (µmhos/cm), $X_4$ = Transparency (cm), $X_5$ = Hardness (mg/l), $X_6$ = Alkalinity (mg/l), $X_7$ = Chloride (mg/l), $X_8$ = Magnesium (mg/l), $X_9$ = Phosphate (mg/l), $X_{10}$ = Nitrate (mg/l), $X_{11}$ = Iron (mg/l), $X_{12}$ = Free $CO_2$ (mg/l), $X_{13}$ = Dissolved oxygen (mg/l), $X_{14}$ = BOD (mg/l), $X_{15}$ = COD (mg/l), $\mu$ = Error component. The regression is computed in SPSS v23.

Shannon-Weiner diversity index (H'), Simpson index (D), Simpson diversity index (E) and Margalef richness index (R) were also calculated as part of this study. The computation was done using PAST v3.26 [17].

## Results

### Analysis of physico-chemical parameters

Observations in this study reveal distinct variations in the values of physico-chemical parameters of Tasek Lake throughout the year (Table 1). The pH lies within a range of 4.85 (pre-monsoon) and 7 (monsoon). Temperature varied from a minimum of 16.87° C in winter to a maximum of 31.21° C in monsoon. Conductivity had a minimum of 17.10 µhos/cm (pre-monsoon) and a maximum of 25.60 µhos/cm (monsoon). Further, transparency varied from 53.87 cm in monsoon to 73.76 cm in winter. Hardness was observed to range within 51.20 mg/l in July to 81.20 mg/l in April. Alkalinity varied from a maximum of 15.99 mg/l (retreating monsoon) to a minimum of 8.49 mg/l (pre-monsoon). The chloride content showed a variation from 7.65 mg/l (winter) to 15.90 mg/l (monsoon). Magnesium also showed a similar variation from 7.73 mg/l (winter) to 15.23 mg/l (monsoon). Phosphate exhibited marginal change in

**Table 1. Physio-chemical parameters of Tasek Lake, 2019 (February)-2020 (January).**

| Parameters | Pre-Monsoon | | | Monsoon | | | | Retreating Monsoon | | Winter | | | Descriptive stat. | |
|---|---|---|---|---|---|---|---|---|---|---|---|---|---|---|
| | Mar | Apr | May | June | July | Aug | Sep | Oct | Nov | Dec | Jan | Feb | Mean | SD (±) |
| pH | 4.85 | 6.43 | 6.89 | 6.79 | 7 | 6.56 | 6 | 6.21 | 6.15 | 5.42 | 5.3 | 5.23 | 6.06 | 1.2 |
| Water (°C) | 21.43 | 23.12 | 26.43 | 27 | 31.21 | 28.97 | 26.83 | 24.9 | 24 | 17 | 16.87 | 17.52 | 23.77 | 4.98 |
| Conductivity (µhos) | 17.6 | 19.1 | 17.1 | 25.1 | 25.6 | 24.1 | 24.6 | 21.6 | 21.85 | 19.6 | 20.1 | 20.6 | 21.41 | 4.38 |
| Transparency (cm) | 63.76 | 64.26 | 64.26 | 58.76 | 53.87 | 56.76 | 60.76 | 69.26 | 63.76 | 71.76 | 70.76 | 73.76 | 70.55 | 7.42 |
| Hardness (mg/L) | 79.2 | 81.2 | 70.2 | 64.2 | 51.2 | 57.2 | 71.4 | 68.2 | 71.2 | 70.2 | 66.2 | 68.2 | 68.21 | 9.08 |
| Alkalinity (mg/L) | 9.49 | 8.49 | 8.99 | 9.09 | 9.49 | 9.99 | 11.99 | 12.49 | 15.99 | 14.49 | 12.49 | 11.99 | 11.19 | 1.82 |
| Chloride (mg/L) | 9.8 | 9.8 | 12.4 | 12.65 | 13.9 | 15.4 | 15.9 | 15.65 | 13.4 | 7.8 | 7.65 | 12.4 | 12.22 | 3.67 |
| Magnesium (mg/L) | 7.73 | 9.13 | 13.73 | 13.98 | 15.23 | 13.33 | 11.73 | 10.98 | 9.73 | 8.73 | 8.23 | 7.73 | 10.85 | 3.8 |
| Phosphate (mg/L) | 2.13 | 2.13 | 2.13 | 2.15 | 2.15 | 2.11 | 2.12 | 2.13 | 2.10 | 2.11 | 2.12 | 2.14 | 2.12 | 2.12 |
| Nitrate (mg/L) | 2.32 | 2.33 | 2.34 | 2.34 | 2.34 | 2.31 | 2.33 | 2.31 | 2.33 | 2.34 | 2.32 | 2.33 | 2.32 | 2.31 |
| Iron (mg/L) | 1.63 | 1.61 | 1.6 | 1.62 | 1.62 | 1.61 | 1.63 | 1.62 | 1.62 | 1.61 | 1.6 | 1.62 | 1.61 | 1.51 |
| Free CO2 (mg/L) | 2.45 | 1.95 | 2 | 2.5 | 3 | 3.5 | 4 | 4 | 3.5 | 3.5 | 3 | 2.5 | 2.99 | 1.57 |
| D.O. (mg/L) | 12.52 | 9.29 | 8.36 | 8.59 | 9.3 | 8.1 | 7.7 | 9.03 | 11.14 | 13.85 | 13.6 | 12.65 | 10.34 | 3.69 |
| BOD (mg/L) | 4.5 | 4.1 | 4 | 3.8 | 3.5 | 3.72 | 4 | 4.4 | 4.45 | 4.5 | 4.6 | 4.5 | 4.17 | 1.85 |
| COD (mg/L) | 23.5 | 24.5 | 24 | 26.5 | 22.5 | 30.5 | 31.5 | 31.75 | 23.5 | 21.5 | 18 | 22 | 24.97 | 5.61 |

values from 2.10 mg/l (retreating monsoon) to 2.16 mg/l (winter). Nitrate concentration exhibited values hovering between 2.31 mg/l and 2.34 mg/l in all seasons. Similarly, values for Iron varied between 1.6 mg/l and 1.63 mg/l. Free $CO_2$ ranged from 2.00 mg/l (pre-monsoon) to 4.00 mg/l (monsoon). For DO, BOD and COD, values ranged from 7.7 mg/l (monsoon) to 13.85 mg/l (winter), 3.50 mg/l (monsoon) to 4.60 mg/l (winter) and 18 mg/l (winter) to 31.75 mg/l (retreating monsoon) respectively.

## Water Quality Index calculation

Table 2 displays the relative weights, BIS standards for drinking water [18] and the WQI value. As part of the calculation, we have chosen only those water quality parameters whose standards are mentioned under BIS. Finally, the water quality has been rated according to its WQI, viz. WQI<50 –"excellent"; 50<WQI<100 –"good"; 100<WQI<200 –"poor"; 200<WQI<300-"very poor" and WQI>300 –"unsuitable for drinking" [14]. The WQI for Tasek Lake is 250.06.

## Abundance and diversity of freshwater plankton

A total of eight classes of algae with varied seasonal distributions were identified as part of the study. Cyanobacteria (Myxophyceae) basically included Anabaena, Coelosphaerium, Micro-cystis, Rivularia, Oscillatoria, Aphanocapsa, Merismopedia and Stigonema. The Cyanophyceae algae comprised of Phormidium, Arthropia, Spirulina, Anacystis, Lyngbya and Gloecocapsa. Bacillariophyceae included Synedra, Amphora, Asterionella, Nitzschia, Cyclotella, Diatoma, Diatomella, Frustulia, Gyrosigma and Meridion. Further, Diatomaceae included Fargilaria, Cymeblla, Amphora and Asterias. Euglenineae comprised of Lepocinclis and Euglena. Desmi-daceae were also found which included Desmidium, Netrium, Cosmariuam and others, followed by the Dinophyceae class. Chrysophyceae included Mallomonas and Synura. Finally, the Chlorophyceae class was composed of at least 20 genus viz., Dinobryon, Chlamydomonas, Chodatelle and so on.

Seasonal abundance for Myxophyceae was highest during March (1422 individuals) compared to a low during August (30 individuals) (Fig 2A and 2B). Dominance of Bacillariophy-ceae was seen during May (2709 individuals) and while the weakest point numbers were observed during November (185 individuals). August witnessed the highest number of Cyano-phyceae at 1508 individuals with the lowest during February at 146 individuals. Chlorophyceae peaked during retreating monsoon (1138 individuals) while falling during pre-monsoon (131 individuals). Dinophyceae numbers were highest during July (944 individuals) and the lowest during December (106 individuals). Further, Chrysophyceae displayed the highest abundance during September (160 individuals) and the lowest during May (88 individuals). Finally, Euglenineae displayed the maximum abundance (160 individuals) during monsoon and the lowest number during pre-monsoon (80).

**Table 2. WQI calculation for Tasek Lake.**

| Parameters | $w_i$ | $W_i$ | C | S | $q_i$ | WQI |
|---|---|---|---|---|---|---|
| pH | 4 | 0.097561 | 6.065 | 6.5 | 39.4225 | 3.846098 |
| Transparency | 2 | 0.04878 | 70.55167 | 30 | 2116.55 | 103.2463 |
| Hardness (mg/L) | 2 | 0.04878 | 68.21667 | 30 | 2046.5 | 99.82927 |
| Chloride (mg/L) | 3 | 0.073171 | 12.22917 | 25 | 305.7292 | 22.37043 |
| Magnesium (mg/L) | 1 | 0.02439 | 10.855 | 30 | 325.65 | 7.942683 |
| Nitrate (mg/L) | 5 | 0.121951 | 2.328333 | 45 | 104.775 | 12.77744 |
| Iron (mg/L) | 3 | 0.073171 | 1.615833 | 0.3 | 0.48475 | 0.03547 |
| Total | 20 | | | | | 250.0477 |

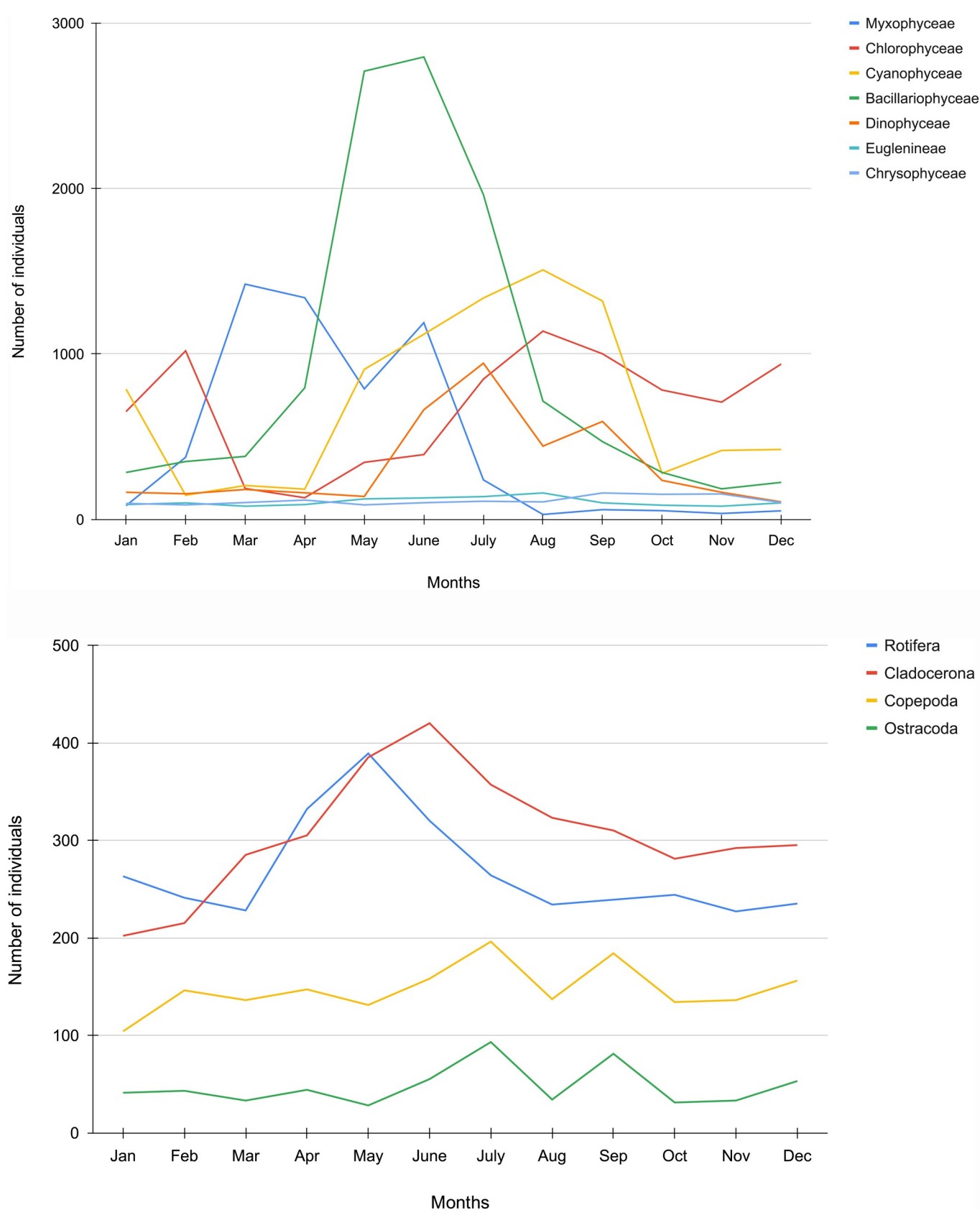

**Fig 2.** a. Monthly variation of phytoplankton population. b. Monthly variation of zooplankton population.

As regards zooplankton, the Copepoda class constituted of Cyclops, Nauplius, Alonella and Canthocampus. Further, Rotifers comprised of Filinia, Trichocera, Schizocera, Asplancha and so on. Cladocera included Daphnia, Moina, Diaphnosoma, etc. Finally, Ostracoda included the species Cypris. Seasonal abundance for Cladocera peaked during June (420 individuals) and fell during January (202 individuals). Copepoda individuals were highest during July (196 individuals) and lowest during January (104 individuals). Ostracoda peaked during July with 93 individuals while fell during October with 31 individuals. Rotifers were most abundant during May (389 individuals) and least during November (227).

Figs 3A, 3B, 3C, 3D, 4A, 4B, 4C and 4D depict the diversity indices of plankton which have been calculated for all the four seasons- winter, pre-monsoon, monsoon and retreating monsoon respectively. All the indices are seen more or less showing higher values during the monsoon season with the dominance of Bacillariophyceae and Chlorophyceae among phytoplankton and, Copepoda and Rotifera among zooplankton.

## Relationship between water quality parameters and plankton abundance

The final models derived from the regression analysis are laid out in expressions (5) and (6).

$$Phytoplankton, Y_1 = 255044.478$$
$$= 205.431X_1 - 436.811X_2 + 637.385X_3 + 182.160X_5 - 1389.703X_6 + 679.840X_7$$
$$- 52364.399X_9 - 32181.472X_{11} + 2590.953X_{12} - 829.914X_{15}(5)$$

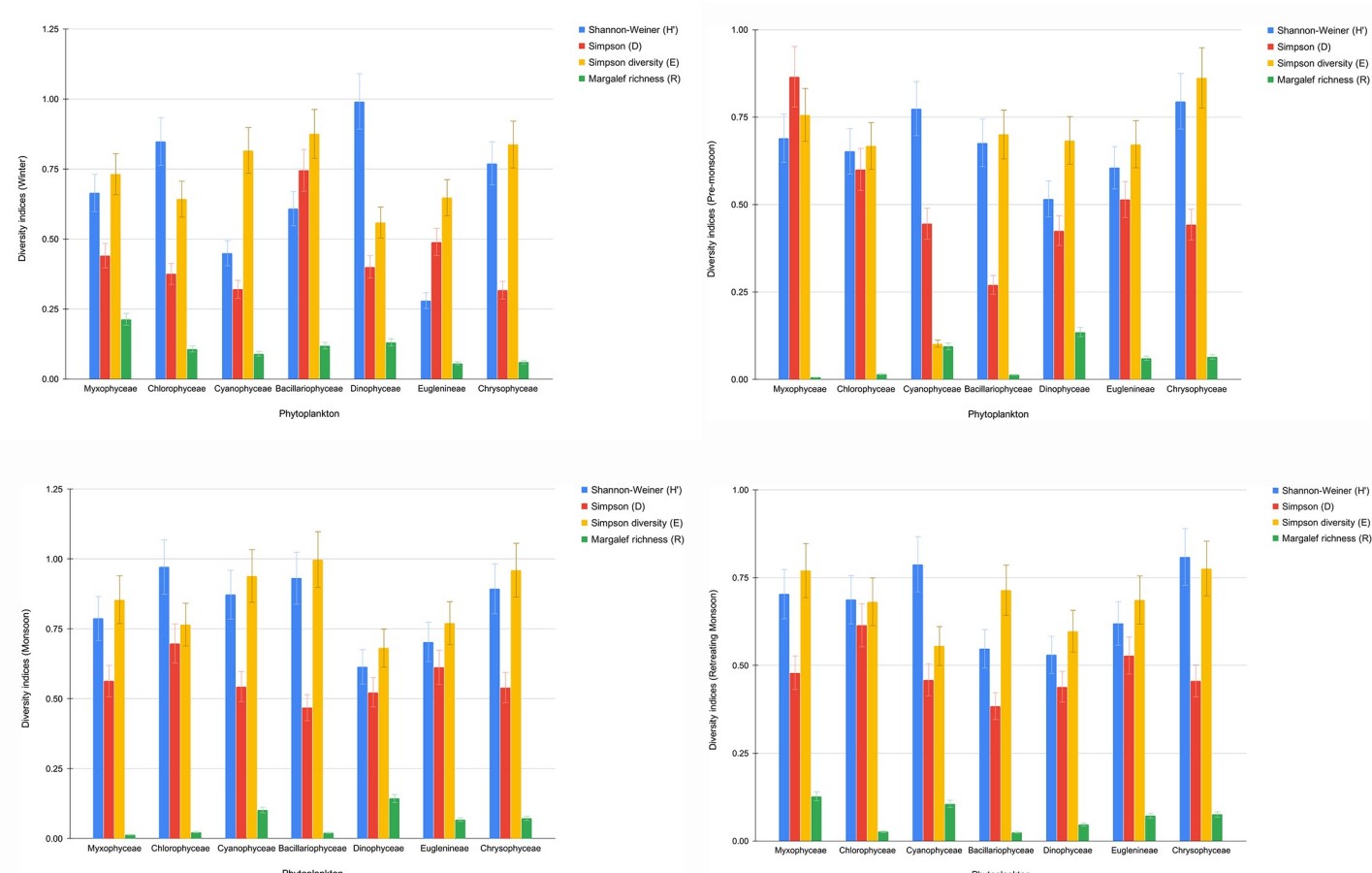

**Fig 3.** a. Diversity indices for phytoplankton (Winter). b. Diversity indices for phytoplankton (Pre-Monsoon). c. Diversity indices for phytoplankton (Monsoon). d. Diversity indices for phytoplankton (Retreating Monsoon).

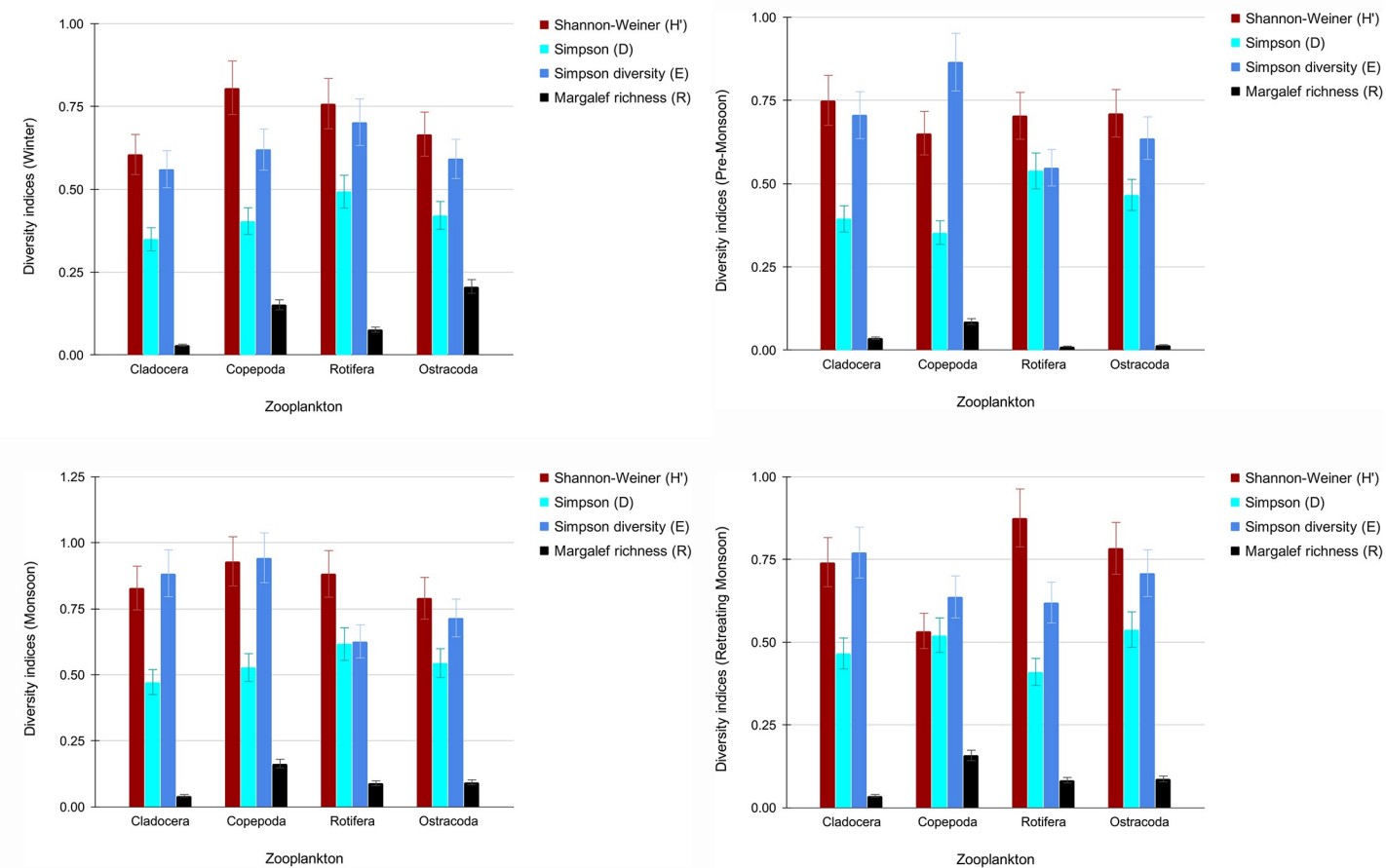

**Fig 4.** a. Diversity indices for zooplankton (Winter). b. Diversity indices for zooplankton (Pre-Monsoon). c. Diversity indices for zooplankton (Monsoon). d. Diversity indices for zooplankton (Retreating Monsoon).

$$R^2 = 0.931 \text{ (Significant)}$$

$$
\begin{aligned}
Zooplankton, Y_2 \\
= 12214.504 + 51.337X_1 - 28.840X_2 + 33.118X_3 + 12.822X_5 - 128.206X_6 + 65.780X_7 \\
- 5772.242X_9 + 2007.169X_{10} - 2152.765X_{11} + 345.238X_{12} - 80.427X_{15}(6)
\end{aligned}
$$

$$R^2 = 0.956 \text{ (Significant)}$$

In the above models, the variables Trans. ($X_4$), Mg ($X_8$), DO ($X_{13}$) and BOD ($X_{14}$) were classified as excluded variables due to their tolerance being null.

## Canonical Correspondence Analysis

The CCA revealed impact of months on the different physio-chemical parameters present in the freshwater Tectonic Lake of the Indo-Burmese Province considered under the present study. Table 3 shows that the physico-chemical parameters are negatively correlated to the environmental changes with respect to different months in a year. Fig 5A shows the relative relationship between the environmental factors (physico-chemical parameters) and the months in a year. The level of hardness was more visible in the months of March and April,

**Table 3. Association between months and parameters.**

| Dimension | Singular Value | Inertia | Chi Square | Sig. (p-value) | Proportion of Inertia | | Confidence Singular Value | |
|---|---|---|---|---|---|---|---|---|
| | | | | | Accounted for | Cumulative | Standard Deviation | Correlation |
| | | | | | | | | 2 |
| 1 | 0.152 | 0.023 | | | 0.556 | 0.556 | 0.018314 | -0.01073 |
| 2 | 0.111 | 0.012 | | | 0.296 | 0.852 | 0.01781 | |
| 3 | 0.056 | 0.003 | | | 0.076 | 0.928 | | |
| 4 | 0.038 | 0.001 | | | 0.036 | 0.964 | | |
| 5 | 0.024 | 0.001 | | | 0.014 | 0.978 | | |
| 6 | 0.022 | 0.000 | | | 0.011 | 0.989 | | |
| 7 | 0.016 | 0.000 | | | 0.006 | 0.996 | | |
| 8 | 0.011 | 0.000 | | | 0.003 | 0.999 | | |
| 9 | 0.006 | 0.000 | | | 0.001 | 1.000 | | |
| 10 | 0.002 | 0.000 | | | 0.000 | 1.000 | | |
| 11 | 0.001 | 0.000 | | | 0.000 | 1.000 | | |
| Total | | 0.041 | 132.491[a] | 0.894 | 1.000 | 1.000 | | |

a. 154 degrees of freedom

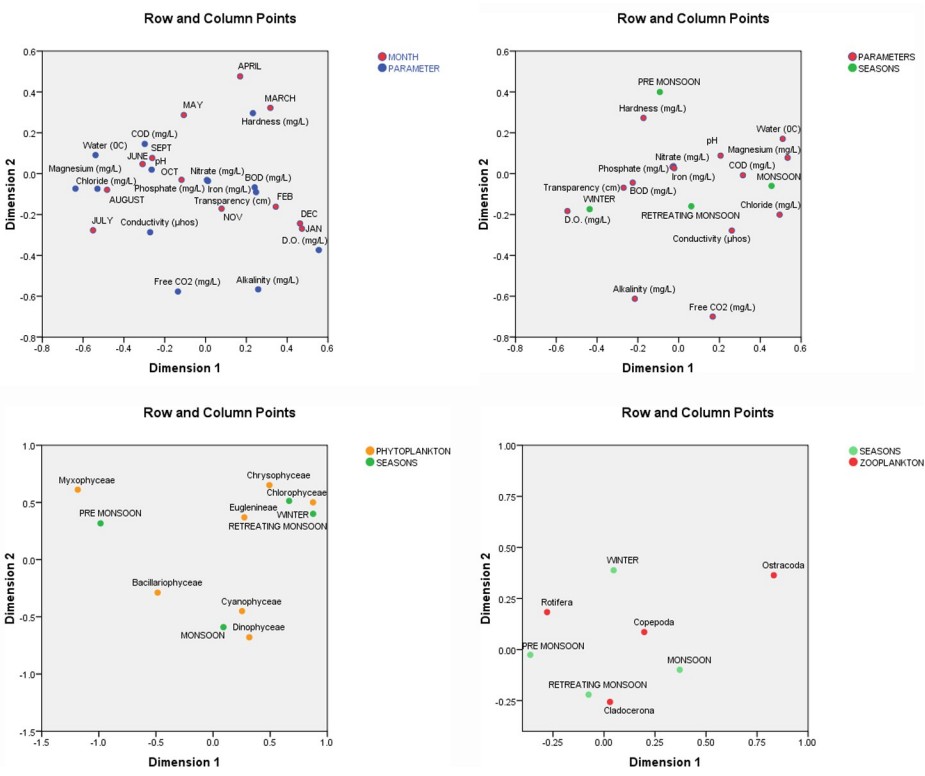

**Fig 5.** a. CCA biplot showing the relationship between environmental parameters and months in a year. b. CCA biplot showing the relationship between environmental parameters and seasons in a year. c. CCA biplot showing relationship between the phytoplankton species and seasons in a year. d. CCA biplot showing relationship between the zooplankton species and seasons in a year.

**Table 4. Association between seasons and physico-chemical parameters.**

| Dimension | Singular Value | Inertia | Chi Square | Sig. | Proportion of Inertia | | Confidence Singular Value | |
|---|---|---|---|---|---|---|---|---|
| | | | | | Accounted for | Cumulative | Standard Deviation | Correlation |
| | | | | | | | | 2 |
| 1 | .517 | .267 | | | .516 | .516 | .026 | .240 |
| 2 | .401 | .161 | | | .311 | .826 | .027 | |
| 3 | .300 | .090 | | | .174 | 1.000 | | |
| Total | | .518 | 537.456 | .000[a] | 1.000 | 1.000 | | |

a. 42 degrees of freedom

while the factors (COD and pH) were highly concentrated in June, September and October. The month of October also observed high level of Phosphate in water. December and January are very closely related with DO, BOD, whereas iron levels were highly concentrated in February. Transparency was closely connected with February as well as with the month November. However, this association between the two sets of multidimensional variables (parameters and months) in our study is found out to be not significant (p>0.05).

The seasonal impact on the environmental factors is very distinct (p<0.05) (Table 4). The concentration level of pH, water (OC) Magnesium and COD were increased in Monsoon while Winter has a negative effect on them (Fig 5B). Retreating Monsoon experienced positive influence on the parameters like Iron, Nitrate Phosphate and Conductivity. High concentration levels of Transparency, BOD and DO were very closely connected to Winter season while during Monsoon pH, water (OC) Magnesium and COD were found to be high in the water samples under our study. Moreover, water quality of the lake was found to be characterized by obvious increasing concentration level of hardness and decreasing concentration level of Alkalinity.

Season is also a significant factor playing a major role in determining the density and occurrence of phytoplankton population in the water body under present study (p<0.05) (Table 5). The Fig 5C shows the relationship between 4 seasons in a year and phytoplankton assemblages at the sampling sites using canonical correlation analysis. The results reveal that Winter and Retreating monsoon accelerate the growth of the phytoplankton species (Chrysophyceae, Chloropohyceae, and Euglenineae). As we can see that Transparency, BOD and DO are very closely connected to winter and Retreating monsoon experienced positive influence on the parameters like Iron, Nitrate Phosphate and Conductivity (Bipolt2), hence the high concentration levels of physico-chemical parameters (Transparency, BOD and DO, Iron, Nitrate Phosphate and Conductivity) can be considered as the accelerators of the above phytoplankton species growth. The growth of Cyanophyceae and dinophyceae is closely associated with the

**Table 5. Association between seasons and phytoplankton species.**

| Dimension | Singular Value | Inertia | Chi Square | Sig. | Proportion of Inertia | | Confidence Singular Value | |
|---|---|---|---|---|---|---|---|---|
| | | | | | Accounted for | Cumulative | Standard Deviation | Correlation |
| | | | | | | | | 2 |
| 1 | .465 | .216 | | | .781 | .781 | .008 | .229 |
| 2 | .236 | .056 | | | .201 | .982 | .009 | |
| 3 | .071 | .005 | | | .018 | 1.000 | | |
| Total | | .277 | 3423.467 | .000[a] | 1.000 | 1.000 | | |

a. 18 degrees of freedom

**Table 6. Association between seasons and zooplankton species.**

| Dimension | Singular Value | Inertia | Chi Square | Sig. | Proportion of Inertia | | Confidence Singular Value | |
|---|---|---|---|---|---|---|---|---|
| | | | | | Accounted for | Cumulative | Standard Deviation | Correlation |
| | | | | | | | | 2 |
| 1 | .076 | .006 | | | .682 | .682 | .018 | -.031 |
| 2 | .047 | .002 | | | .264 | .946 | .018 | |
| 3 | .021 | .000 | | | .054 | 1.000 | | |
| Total | | .008 | 25.402 | .003[a] | 1.000 | 1.000 | | |

a. 9 degrees of freedom

season Monsoon and the concentration levels of pH, water (OC) Magnesium and COD are found to be very high during monsoon, thus these are the primary environmental factors affecting the phytoplankton species growing in Monsoon.

The result significantly shows that the abundance of zooplankton species is negatively or positively affected by the seasonal environmental effects (Table 6). Clearly, the growth of the zooplankton species located in the first quadrant and fourth quadrant of the trigonometric circle are affected strongly and negatively by the season located in the third quadrant (Retreating Monsoon) (Fig 5D). This implies that the increase in concentration of the physico-chemical parameters closely associated with Retreating Monsoon viz. Iron, Nitrate Phosphate and Conductivity restrains the growth and proliferation of these species. On the other hand, the growth of zooplankton species (Ostracoda, Copepoda and Cladocerona) is found to be positively characterized by the high concentration level of Transparency, BOD DO, pH, water (OC) Magnesium and COD.

## Ichthyofaunal diversity

Table 7 depicts the ichthyofaunal diversity of Tasek Lake wherein 16 families and 51 species belonging to 10 orders have been identified. Major orders include Cypriniformes, Channiformes and Siluriformes which further include major fish species such as the Chocolate Mahseer (*Neolissochilus hexagonolepis*), Golden Mahseer (*Tor putitora*), *Cirrhina mrigala*, *Notopterus chitala*, Rohu (*Labeo rohita*) and so on.

## Discussion

### Interpreting the hydrobiology of Tasek Lake

Assessing the ecological health of a water body is essential for effectively managing the quality of water as well as sustaining pristine aquatic ecosystems. Anthropogenic interference in freshwater river systems have increased rapidly since urbanisation has reached nook and corners of the globe [19]. This applies appropriately for Tasek Lake. Earlier a quiet spot with clean environment, currently the Lake is visited by numerous tourists arriving from all over the world. Such anthropogenic content is bound to have a detrimental effect on its water quality and overall ecosystem. Such degradation will, in turn, damage the lake's substantial fish resources potential [20].

Results of the present study demonstrate that the pH level of Tasek Lake is a minimum of 4.85. This is below the minimum BIS standard limit of 6.5 [18]. Notably, pH balance is critical for overall well-being of ecosystems [20]. These levels affect the behaviour of organisms as well as their osmo-regulatory and respiratory functions. Acidic water tends to make itself corrosive and thus, poses threats to human consumption. The level recorded in our study may be

**Table 7. Ichthyofaunal diversity of Tasek Lake.**

| Order | Family | Sub-family | Species | |
|---|---|---|---|---|
| Clupeiformes | Clupeidae | | *Gudusia chapra* | Hamilton, 1822 |
| Osteoglossiformes | Notopteridae | | *Notopterus chitala* | Hamilton, 1822 |
| Cypriniformes | Cyprinidae | Abramidinae | *Salmostoma bacaila* | Hamilton, 1822 |
| | | Rasborinae | *Barilius bendelisis* | Hamilton, 1822 |
| | | | *Barilius barna* | Hamilton, 1822 |
| | | | *Barilius barila* | Hamilton, 1822 |
| | | | *Danio (Danio) aequipinnatus* | McClelland, 1839 |
| | | | *Danio (Brachydanio) rerio* | Hamilton, 1822 |
| | | | *Rasbora elanga* | Hamilton, 1822 |
| | | | *Esomus danricus* | Hamilton, 1822 |
| | | | *Barilius bendelisis* | Hamilton, 1822 |
| | | Cyprininae | *Tor tor* | Hamilton, 1822 |
| | | | *Chagunius chagunio* | Hamilton, 1822 |
| | | | *Puntius clavatus* | McClelland, 1839 |
| | | | *Labeo boga* | Hamilton, 1822 |
| | | | *Cirrhina reba* | Hamilton, 1822 |
| | | | *Labeo pangusia* | Hamilton, 1822 |
| | | | *Accrossocheilius hexagonolepis* | McClelland, 1839 |
| | | | *Labeo calbasu* | Hamilton, 1822 |
| | | | *Tor putitora* | Hamilton, 1822 |
| | | | *Osteobrama cotio cotio* | Hamilton, 1822 |
| | | | *Puntius chola* | Hamilton, 1822 |
| | | | *Labeo rohita* | Hamilton, 1822 |
| | | | *Garra nasuta* | McClelland, 1839 |
| | | | *Crossocheilus latius latius* | Hamilton, 1822 |
| | | | *Labeo dero* | Hamilton, 1822 |
| | | | *Cirrhina mrigala* | Hamilton, 1822 |
| | Cobitidae | | *Botia rostrate* | Gunther, 1868 |
| | | | *Lepidocephalichthys guntea* | Hamilton, 1822 |
| | | | *Botia histrionic* | Hamilton, 1822 |
| | Psilorhynchidae | | *Psilorhynchus balitora* | Hamilton, 1822 |
| Tetraodontiformes | Tetraodontidae | | *Tetraodon cutcutia* | Hamilton, 1822 |
| Channiformes | Channidae | | *Channa orientalis* | Hamilton, 1822 |
| | | | *Channa punctate* | Bloch, 1793 |
| | | | *Channa striatus* | Hamilton, 1822 |
| | | | *Channa marulius* | Hamilton, 1822 |
| Gobiiformes | Gobidae | | *Glossogobius giuris* | Hamilton, 1822 |
| Anabantiformes | Nandidae | | *Nandus nandus* | Hamilton, 1822 |
| | | | *Anabas testudineus* | Hamilton, 1822 |
| Siluriformes | Amblycipitidae | | *Amblyceps mangois* | Hamilton, 1822 |
| | Bagridae | | *Aorichthys seenghala* | Hamilton, 1822 |
| | | | *Mystus cavasius* | Hamilton, 1822 |
| | | | *Sperata aor* | Hamilton, 1822 |
| | | | *Mystus vittatus* | Bloch, 1794 |
| | | | *Mystus tengara* | Hamilton, 1822 |
| | Siluridae | | *Wallago attu* | Bloch & Schneider, 1801 |
| | | | *Ompok pabo* | Hamilton, 1822 |
| | Heteropneuestidae | | *Heteropneustus fossilis* | Bloch, 1794 |

*(Continued)*

**Table 7.** (Continued)

| Order | Family | Sub-family | Species | |
|---|---|---|---|---|
| Perciformes | Chandidae | | *Chanda ranga* | Hamilton, 1822 |
| | Centropomidae | | *Chanda nama* | Hamilton, 1822 |
| Synbranchiformes | Mastacembelidae | | *Mastacembalus armatus armatus* | Lacepède, 1800 |

attributed to high tourist influx in the lake and resultant polluting activities [21]. Besides, the temperature level of Tasek Lake is found to be within acceptable limits. However, warming of the lake water is noticed if we consider previous studies [8]. This may be due to a cumulative warming of the region's climate as well as its groundwater temperature [22].

The underlying importance of conductivity lies with the amount of substances dissolved in water. These substances provide fillip to electric flow. Higher the conductivity of water, more are the substances/chemicals present in it and worse is the water quality. Tasek Lake presents a moderate conductivity peak of 25.6 μhos/cm during July (monsoon). This fortunately falls within international standards. It is also lesser than Dal Lake that has a conductivity range of 100–565 μhos/cm [23], Loktak Lake [24] and Chandubi Lake [25] Similarly, transparency of water is a measure of water quality as it indicates the depth to which light can penetrate water. Accumulation of dissolved solids can reduce the transparency of water. Barrier to sunlight can substantially inhibit growth of underwater plants and throw the aquatic ecosystem off-balance [26]. Heavy rains naturally cause water to become cloudy. However, pollution from anthropogenic sources is a major cause too. Tasek Lake has a minimum transparency of 53.76 cm, witnessed in monsoon. This may be attributed to heavy rains as well as some pollution due to tourist influx. Monsoon is the time when Meghalaya experiences a high number of visitors.

The hardness of water refers to its high mineral content. In Tasek Lake, the hardness seemed to decrease mainly during the monsoon season while rising during pre-monsoon. This may be due to high evaporation rates during pre-monsoon or the lack of freshwater addition. As per [27], Tasek Lake falls between the soft (0–60) and moderately hard category (61–120). Compared with lakes such as Macferson Lake in Allahabad [28] or the Velachery Lake in Chennai [29], the hardness of Tasek's water is less. But in general terms, such hardness is undesirable given its surrounding pristine environment. Comparing urban lakes parameters to lakes as Tasek, thus, help us visualise the current water quality situation clearly. On the other hand, alkalinity is another important marker of aquatic productivity. Tasek's water chemistry consists of marginal variations in the alkalinity parameter. A peak value is seen during retreating monsoon. This parameter lies within acceptable standards and hence, is of not much concern. It should be noted that water with low alkalinity levels, tend to be more corrosive [30].

Chloride concentration is good indicator of organic pollution as it is highly toxic even if alone ($Cl_2$) [31]. In Tasek Lake, the chloride content has a maximum of 15.50 mg/l which is much less than BIS standards of 250 mg/l. The existing content may be traced to chlorides originating from rocks or agricultural run-off. Since, there are no heavy industries or highly urbanised settlements in vicinity, the chloride level is acceptable. However, the values are higher than those observed previously [32]. Besides, the maximum magnesium and phosphate concentrations are seen during the monsoon season. The values for these parameters are well within prescribed drinking water limits. Nitrate and iron concentration display an even distribution throughout the year. High nitrate levels are normally indicators of agricultural runoff, sewage disposal or other anthropogenic interventions [33]. However, the nitrate levels are lower than Manipur's Loktak Lake (93.67 mg/l) but are marginally higher than those witnessed

in Deepor Beel (0.11 mg/l), a freshwater Ramsar Site located in the bordering State of Assam [34]. The latter is exposed to moderate pollution levels invariably indicating similar pollution levels in Tasek Lake. Nevertheless, the observations of physico-chemical parameters in our study is consolidated by the WQI result which revealed a value of 250.06, indicating the water quality to be "very poor". To place this in perspective, Tasek's water is better than that of Kashmir's Dal Lake whose WQI ranges between 324.97 and 4130.99 [35] while is worse than Karnataka's Hebbal Lake that has a WQI range of 59.80–136.09 [36].

Our study of Tasek's plankton abundance revealed another set of data which if related with physico-chemical parameters, may lead to a viable conclusion about its overall ecosystem health. Among the phytoplankton, Bacillariophyceae had the highest number of individuals in all seasons throughout the year with dominance in pre-monsoon, gradually declining towards retreating monsoon. Similar dominance was shown by Chlorophyceae and Cyanophyceae. The least abundance was observed for Chrysophyceae which corroborates findings of [19]. In case of zooplankton, Cladocera displayed a more or less persistent abundance during all seasons while Ostracoda showed the least.

Interpreting the effect of physico-chemical parameters on plankton abundance, an analysis of the regression model results is needed. The results clearly indicate that these parameters tend the effect plankton abundance. For instance, the negative effect of low pH on plankton growth is well known [37]. We saw during the study that at the same time when pH of water was high (monsoon), both phytoplankton and zooplankton growth rate had also increased. This is further revealed by the regression models where the relationship with pH is seen as positive. Similarly, warming of the water induces plankton growth as well [38]. Notably, the growth of Cholorophyta and Cyanobacteria as well as Cladocera is conducive to higher temperatures. The regression model indicates the same with a positive relation.

Conductivity is observed to have varied effects on plankton abundance. While Cyanobacteria abundance is conventionally associated with higher conductivity (possibility of nitrogen fixation), in other cases no such significant relationships are seen. Nevertheless, the source of conductivity may be attributed to nutrient and mineral sources which may explain the positive relationship seen in our regression model. Similar conclusions can be for the hardness parameter as we can it having a positive relationship with plankton abundance. Alkalinity, on the other hand, can be seen sharing a negative relationship with plankton abundance [39] notes that high alkalinity results in lowering of pH and a low pH inhibits plankton growth while reducing plankton biomass, species number and diversity. Overall, high alkalinity can damage the planktonic structure of a water body. Observations in our study depict a similar picture. Alkalinity seems to decrease by monsoon wherein pH rises. Therefore, higher planktonic abundance can be seen at that time. Notably, the levels of alkalinity or planktonic growth have increased over the years [40] and hence, points toward artificial interference in the lake's water chemistry. This is corroborated by the seasonal trends of diversity indices as well.

Higher chloride concentration in water is considered detrimental to plankton growth. Husnah and Lin [41] state that an increase in chloride concentration by 0.25 mg/l can reduce primary productivity by more than 50% while killing more than 40% of zooplankton populations. The negative relationship obtained in our regression model verifies this. Tasek Lake's chloride levels have increased over the years which directly indicate organic pollution but given its comparatively low levels, the effect on plankton population may be disregarded at present. In contrast, nitrate, phosphate and iron are crucial factors for boosting plankton growth [42]. At the same time, these minerals are indictors of freshwater contamination. Although, their levels in Tasek Lake are well within acceptable limits, growth in the near future is not unthinkable. They showcase a positive relationship with plankton abundance in the regression model corroborating contemporary evidence. Notably, high level of nitrates and phosphates have been

found in Loktak Lake and Dal Lake, which have a similar environmental backdrop as Tasek Lake [24]. Besides, algal blooms in iron-rich water bodies are commonly seen. As such, abundance of these elements may lead to excessive algal blooms, thereby damaging the water quality of the lake. It is imperative to note here that fishery potential of the water body will substantially decrease as a result of this. Similar conclusions can be derived from the levels of DO, BOD and COD. Although within acceptable limits, low DO and high BOD or COD has detrimental effects on the plankton growth of a water body. Oxygen depletion is known to cause stress and untimely death of aquatic biota [43]. This inhibits aquaculture [33] in any water body and requires effective ecological restoration as understood from the experience in Dal Lake. Notably, high plankton growth can lead to decrease in DO, resulting in high free carbon in the lake's water. CCA corroborates these results as we have seen that physico-chemical parameters are negatively related to the different months while the seasonal impact is very distinct. It is seen that seasons have significant effect on planktonic assemblages. Since, difference in characteristics of physico-chemical parameters can be seen to change in different seasons, we can suspect that adverse changes in the environmental factors can definitively impact the planktonic assemblages.

The trend of diversity indices clearly indicates that plankton diversity, in general, responds to the change in physico-chemical parameters of Tasek's water. It is well known that plankton diversity corresponds to nutrient changes in water [44]. High nutrient influx contributes to algal blooms. Tasek Lake boasts of high plankton diversity, spread likewise over all the seasons. Margalef richness index displays moderate species richness as well. Therefore, Tasek Lake is characterised by a distinct eutrophic status which is in contrast with past literature. This calls for its urgent ecological restoration of the Lake. Failure to do so will lead to slow destruction of the immense fishery potential as well as pristine aquatic ecosystem of this major tectonic lake.

The ichthyofaunal diversity of Tasek Lake gives us a significant insight into the lake's fishery potential. For instance, the Mahseers have immense commercial, sport and table value. The fish is very popular among the local fishermen due to its tough texture, large size and longer shelf life [45]. Endemic to Asian waters, Mahseers are highly sought for recreational fisheries due to its popularity amongst anglers. Besides, the importance of Rohu is undeniable due to its potential for high growth and consumer preferences. It is considered as one of the most important freshwater fish species in South Asia. Similarly, Chitala has commercial value in terms of consumer preferences, aquariums and sport. It is highly appreciated as a fishery resource in India as well as Southeast Asia.

The water quality analysis in this study indicates that there is high imminent threat to the existence of these freshwater fishes. Temporal adverse variations in the physico-chemical parameters, as seen in this study, may result in adverse on towards fish diversity. For instance, acidic pH can retard fish growth substantially. Moreover, a spike in unfavourable conditions will lead to plankton growth which will eventually decrease DO thereby affecting the fish population. High concentrations of nitrate and chlorine can prove highly toxic to freshwater fishes. Effects pertain to adverse morphological and histopathological changes [46]. Similarly, elevated phosphate levels may lead to algal blooms which can eventually deplete oxygen in the water [47]. Ultimately, an imbalance or elevation in the typical physico-chemical parameters or mineral content in the lake can substantially harm the fish potential of Tasek Lake. This calls for conservation of its rich fish resources.

Pisciculture in Tasek Lake would require captive breeding and artificial propagation of certain high value fishes such as Mahseers. Other mahseer species might include *Tor tambroides* and *Tor khudree*. Efforts should be made to use seed production technology and operative induced breeding techniques that go beyond traditional fish farming techniques but are simple to implement at the same time. Proper monitoring of before and after stocking of fish

population would have to be done to ensure the techniques are effectively implemented. Further, introducing fingerlings of high value freshwater fishes such as Grass carp (*Ctenopharyngodon idella*) or Silver carp (*Hypophthalmichthys molitrix*) fishes could also go a long way in boosting the fishery potential of the lake.

## Economic potential of pisciculture in Tasek Lake

In the 21st century, fisheries are a vital source of food security and an important avenue for creation of employment. Rural-urban migration is greatly contained through creation of local employment opportunities and upliftment of the local economy. The issue of food security pertains from the fact that fishes are a rich source of macro- and micro-nutrients, namely, iron, fats, carbohydrates, lipids [48]. Fish oil is considered highly beneficial for the heart. Keeping this at the centre, large markets have cropped up all over the world that trade in fish-based food products and health supplements. In 2019, the global omega-3 product market was valued at USD 2.50 billion. Overall, with 177.8 million tonnes of trade in fish and seafood in January 2020, the global market size is expected to reach a value of USD 155.20 billion by 2023. Moreover, India ranks third in the list of global fish exporters, after China and Indonesia. This invariably indicates the necessity of developing fisheries across all freshwater water resources in India, wherever the geographical and physiological conditions are conducive for doing so.

Meghalaya is largely a fish consuming state which also practices aquaculture on a significant scale [20]. Its freshwater resources are conducive to extensive fish farming. Traditionally, fish practice is done for recreational purposes by the Khasi, Garo and Jaintia, which are predominant tribes of the region [47]. Chocolate mahseers are mostly popular amongst them. For a small fee, local entrepreneurs have started opening up their own ponds and reservoirs to anglers for fishing. Competitions can be seen organised in various parts of the state wherein hundreds of anglers take part. Moreover, Meghalaya is known to receive the highest rainfall in earth. This gives ample scope for developing the fishery potential of its indigenous natural lakes.

While studying the water chemistry of Tasek Lake, frequent visits were made to the nearby villages to gather information on whether any plan has been proposed to develop Tasek Lake into a major fishery. Our primary investigation revealed that the local authorities were planning since long to develop a local fishery which have not been realised yet. Same has been the situation with nearby freshwater lakes, Wari and Dachi Lake. A reason for doing so is the large tourism potential of the lake. Tasek is visited by hundreds of people throughout the year. Tickets for boating and fishing range from INR 10–50. This local industry is managed by a group of local youths belonging to nearby villages. As such, establishing a fishery is bound to hamper the tourism potential of the lake. Further, heavy tourism is detrimental to the lake's long-term ecological sustainability and well-being of its aquatic biota. Moderate eutrophication, as indicated by results of this study, can be visibly witnessed at any time of the year. Therefore, strategies for converting the Tasek Lake, at least partially, is pertinent to redeem the pristine quality of its water. A cost-benefit analysis is necessary before moving forward in this direction.

The benefits associated with converting Tasek Lake into a fishery pertain to livelihood creation, increase in per capita income of the locality and per capita nutrient consumption of the households [49]. Besides, ecological restoration of the lake will reveal substantial benefits in terms of recreational opportunities and aesthetics. It will also contribute to Meghalaya's total fish exports which has declined significantly in recent years. Interestingly, the state imports most of its consumed fish. In an ideal situation, this should not have been the case. Nevertheless, such steps would enhance the overall food security of the region by providing a cheap source of nutrients to the tribal and rural people.

On the other hand, costs would essentially include revenue foregone due to reduced tourist footfalls. However, this is negligible compared to the overall financial benefits earned through fish production, consumption and export. Reducing the costs would require breaking any local nexus that find it beneficial to earn money through tourism activities. At this juncture, the concept of recreational fisheries can be thought of which is quite popular in various developed and developing nations [50]. It is considered a part of their culture as well as socio-economic structure. The practice involves a catch and release process, where a required amount of fishes is re-released into the lake for regeneration purposes. This helps sustain the lake's ecology whilst also bringing in income through recreational activities [51]. Therefore, as part of this study, we recommend a model of recreational fishery to be built around Tasek Lake. The management of this would lie with the local government which will work in tandem with the neighbouring villages. The revenue earned should be shared by both in an 80:20 ratio- 80% for villagers and 20% for government. Implementing this will definitely uplift the local economy while maintaining the ecology of Tasek Lake.

## Conclusions

The present study involved the limnological analysis of Tasek Lake, a tectonic lake of the Indo-Burma Province, with respect to its physico-chemical parameters and plankton abundance over four seasons. The WQI value obtained is 250.06 concludes that the water falls under "very poor" category. Further, seasonal plankton abundance was recorded and plankton diversity was calculated using the Shannon-Weiner diversity index, Simpson index, Simpson diversity index and Margalef richness index. The seasonal trend of physico-chemical parameters and plankton abundance indicate moderate eutrophication, indicating imminent ecological degradation of the lake. This is further established by the WQI and plankton diversity indices. CCA was also conducted to assess the relationship between seasons and planktonic assemblages. We have also assessed the fishery potential of the lake by recording the abundance of its ichthyofauna. 16 families and 51 species belonging to 10 orders were identified. Findings necessitate effective and urgent ecological restoration of the lake. Besides, as part of the brief economic analysis of the fishery potential of the lake, it is recommended that the lake be converted into a recreational fishery. This model is widespread in USA and China. With the ecology of pristine ecosystems quickly moving into oblivion due to anthropogenic destruction, it has become imperative that ecologically rich freshwater lakes as Tasek Lake is brought back from the brink of annihilation. This requires restoration to a state where the lake's ecosystem health is sustained while its ecological potential is successfully harnessed, reflecting the move towards an environmentally sustainable development process.

## Author Contributions

**Conceptualization:** Arup Kumar Hazarika, Unmilan Kalita.

**Data curation:** Arup Kumar Hazarika, Unmilan Kalita.

**Formal analysis:** Arup Kumar Hazarika, Unmilan Kalita, Dulumoni Das.

**Investigation:** Arup Kumar Hazarika, Unmilan Kalita.

**Methodology:** Arup Kumar Hazarika, Unmilan Kalita.

**Project administration:** Arup Kumar Hazarika, Unmilan Kalita.

**Resources:** Arup Kumar Hazarika, Unmilan Kalita.

**Software:** Arup Kumar Hazarika, Unmilan Kalita, Dulumoni Das.

**Supervision:** Arup Kumar Hazarika.

**Validation:** Arup Kumar Hazarika, Unmilan Kalita, Dulumoni Das.

**Visualization:** Arup Kumar Hazarika, Unmilan Kalita.

**Writing – original draft:** Arup Kumar Hazarika, Unmilan Kalita, Rev. George Michael.

**Writing – review & editing:** Arup Kumar Hazarika, Unmilan Kalita, Rev. George Michael, Saroj Panthi, Dulumoni Das.

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
