## [Decision Letter · Decision Letter 0]

21 Sep 2020

PONE-D-20-26442

Hydrobiology of a Freshwater Tectonic Lake of the Indo-Burmese Province

PLOS ONE

Dear Dr. Panthi,

Thank you for submitting your manuscript to PLOS ONE. After careful consideration, we feel that it has merit but does not fully meet PLOS ONE’s publication criteria as it currently stands. Therefore, we invite you to submit a revised version of the manuscript that addresses the points raised during the review process.

We look forward to receiving your revised manuscript.

Kind regards,

SSS Sarma

Academic Editor

PLOS ONE

Additional Editor Comments:

Dear Authors

Three reviewers have recommended your contribution with some corrections. I therefore encourage you to submit a revised version for re-reviewing.

Sincerely

SSS Sarma

Handling Editor

Journal Requirements:

5. Please ensure that you refer to Figure 5 in your text as, if accepted, production will need this reference to link the reader to the figure.

Reviewers' comments:

Reviewer's Responses to Questions

**Comments to the Author**

1. Is the manuscript technically sound, and do the data support the conclusions?

Reviewer #1: No

Reviewer #2: Partly

Reviewer #3: Yes

2. Has the statistical analysis been performed appropriately and rigorously? 

Reviewer #1: I Don't Know

Reviewer #2: Yes

Reviewer #3: Yes

3. Have the authors made all data underlying the findings in their manuscript fully available?

Reviewer #1: No

Reviewer #2: Yes

Reviewer #3: Yes

4. Is the manuscript presented in an intelligible fashion and written in standard English?

Reviewer #1: No

Reviewer #2: No

Reviewer #3: Yes

5. Review Comments to the Author

Reviewer #1: Line 74 Reference required for origin of the lake

Study area: details needed about depth (mean & max); inflows, outflows; climate of the area

Sample Collection (lines 79-87): provide information on the location and number of sampling sites in the lake;

samples taken from surface or at some depth; method of sampling (manual/sampler)

APHA describes several methods for each parameter: describe which of the methods were used

Data are provided in the mss for every month but this is not mentioned here that sampling was done at monthly intervals.

The WQI was calculated and interpreted in context of drinking water standards. How is that relevant to plankton and fishery development? How were the diversity indices calculated. some references are required.

Any water body could be converted for fisheries but the relationship between this study and the recommendation is not clear.

Will the fisheries protect the current fish diversity or promote a few species at the cost of most of them?

What prompted the authors to suggest this, and did they consider the production potential without additional inputs?

Fig 1 has no coordinates, no mark of sampling locations and directions; hence of no value.

Figure 2a, 2b have no Y axis with value and unit, and hence are wholly hypothetical and unacceptable.

Fig 3a to 3d, and 4a to 4d: Y axis is not identified, what are the values? What are the letter symbols?

In my view that tables 3-6 presenting association analysis are not required, or may be included in the supplementary material.

Table 7 on fish diversity can reduced to a simple list of species.

The language and style of presentation need much improvement.

I do not consider the manuscript suitable for publication.

Reviewer #2: Introduction

Goals are somewhat confusing; they need modification. Though field works generally lack hypothesis, I wonder if authors can make one.

Material and Methods

Much of this section is fragmented and redundancy.

Details such as frequency procedures, sampling period, and number of replicates of each parameter are missing. Missing Literature for fish community identification.

Results / Discussion

Overall, results should be described in terms of trends. Incomplete and redundant descriptions. Without highlighting the essence. It is necessary to highlight the effect of the interactions precisely, or statistically. In the graphs, the mean and the standard error of how many replicates ? Graphics need description of their symbols, or footnote. Taxonomic groups must be in italics. Several misspellings in phytoplankton scientific genera names. Redundant descriptions for similar statements Pages 120-121, 175-177

Which is the monsoon period –months ?? What are the meteorologial conditions during this period.

With only slighly variation, how retreating Monsoon has positive influence on the parameters like Iron, Nitrate Phosphate and Conductivity ?? In biological terms what does that mean. Pages 166-168, 237-238

The growth of zooplankton species (Ostracoda, Copepoda and Cladocerona) is found to be positively characterized by the high concentration level of Transparency, BOD DO, pH, water (OC) Magnesium and COD. Are these are all indirect rather than direct interactions Clarify so.

Transparency of water is a measure of water quality as it indicates the depth to which light can penetrate water. Accumulation of dissolved solids can reduce the transparency of water.’’ Pages 296-298. Also ‘’ This may be attributed to heavy rains as well.. pages 301-303

Among the phytoplankton, Bacillariophyceae had the highest number of individuals in all seasons throughout the year with dominance in pre-monsoon, gradually declining towards retreating monsoon. It seems that these groups may represent the typical dynamics induced by silicates together with pH, nutrients (P, N) sources. Does this tectonic aquatic system exhibit a particular mineral physico-chemical environment ??

In order to see a more clear interaction between the plankton groups and the ichthyofaunal diversity, authors must define fish feeding grouping-structure in the lake.

Paraphrasing of the results here is needed. Unfounded arguments and speculation, in general.

Where, can be seen this statement ‘’ Margalef richness index displays moderate species richness as well’’ ?? Pages 390-391

How is that the diversity indices and the environmental conditions are associated with fisheries potential. Explain or delete it.

Unnecessary over citation and outdated literature. As far as possible cite only indexed articles. Popular articles, news paper statements must be avoided and they are rigorously evaluated by peer review processes

Reviewer #3: The manuscript is well written and I congratulate the authors for their future work also. I request the authors go through line no 134, and correct the value of filtered through plankton net.

1. Line 134 – how much liter of water filtered through net?

I recommend the paper for publication after the minor correction.

6. PLOS authors have the option to publish the peer review history of their article (what does this mean?). If published, this will include your full peer review and any attached files.

Reviewer #1: No

Reviewer #2: No

Reviewer #3: No

---

## [Author Response · Author response to Decision Letter 0]

26 Sep 2020

Dear Sir/Mam,

We have revised manuscript according to the suggestion / comments of reviewers.

---

## [Editor Report · Decision Letter 1]

1 Oct 2020

Ecological Status of a Freshwater Tectonic Lake of the Indo-Burmese Province: implications for livelihood development

PONE-D-20-26442R1

Dear Dr. Panthi,

We’re pleased to inform you that your manuscript has been judged scientifically suitable for publication and will be formally accepted for publication once it meets all outstanding technical requirements.

Kind regards,

SSS Sarma

Academic Editor

PLOS ONE
---

## [Editor Report · Acceptance letter]

22 Oct 2020

PONE-D-20-26442R1 

Ecological status of a freshwater tectonic lake of the Indo-Burmese province: implications for livelihood development 

Dear Dr. Panthi:

I'm pleased to inform you that your manuscript has been deemed suitable for publication in PLOS ONE. Congratulations! Your manuscript is now with our production department. 

Kind regards, 

on behalf of

Professor SSS Sarma 

Academic Editor

PLOS ONE